# Materials and Energy Balance of E-Waste Smelting—An Industrial Case Study in China

Fengchun Ye [1,2], Zhihong Liu [1,3] and Longgong Xia [1,*]

[1] Refractory Material Metallurgy and Slag Chemistry Research Group, School of Metallurgy and Environment, Central South University, Changsha 410083, China; yefengchun@nerin.com (F.Y.); zhliu@csu.edu.cn (Z.L.)
[2] Jiangxi Huanan Nerin Precious Metal Technology Co., Ltd., Fengcheng 331100, China
[3] National Engineering Research Center for Low Carbon Nonferrous Metallurgy, Central South University, Changsha 410083, China
[*] Correspondence: longgong.xia@csu.edu.cn; Tel.: +86-15874957018

**Abstract:** The application of Nerin Recycling Technologies (NRT) in electronic waste (E-waste) smelting was introduced in this study, and the material and energy balance was calculated based on the practical data with the METSIM software (METSIM International, USA). The main results are as follows: (1) the optimized processing parameters in the NRT smelting practice were the E-waste feeding rate of 5.95 t/h, oxidation smelting duration of 3.5 h, reduction smelting duration of 0.5 h, oxygen enrichment of 21–40 vol.%, oxygen consumption of 68.06 $Nm^3$/ton raw material, slag temperature of 1280 °C, slag composition: $Fe/SiO_2$ mass ratio of 0.8–1.4, CaO, 15–20 wt.%, Cu in crude copper $\geq$ 95 wt.%, Cu in slag, 0.5 wt.%, recovery of Cu, Au, and Ag $\geq$ 98%; (2) 98.49% Au, 98.04% Ag, 94.11% Ni, and 79.13% Sn entered the crude copper phase in the smelting process, 76.73% Pb and 67.22% Zn volatilized to the dust phase, and all halogen elements terminated in the dust and off-gas; (3) total heat input of the process was 79,480 MJ/h, the energy released by chemical reactions accounted for 69.94% of the total, and heat from fuels burning accounted for 33.04%. The energy brought away by the off-gas was 38,440 MJ/h, which was the largest part in heat output. The heat loss with the smelting slag accounted for 28.47% of the total.

**Keywords:** secondary copper; circular economy; decarbonization; dioxin; halogen

## 1. Introduction

Metal production from secondary resources, such as urban mining, industrial waste, etc., has the advantages of low energy consumption, low $CO_2$ emission, and integrated economic and environmental benefits, etc., and it can also address the problem of resource depletion [1–3]. Electronic waste (E-waste) is a typical kind of resources in urban mining; about 53.6 million tons of E-waste were produced in 2019; it is also one of the fastest growing solid wastes in the world [4]. The global average annual growth rate of E-waste was 4.6% in the past five years, which is 1.8 times larger than that of the global GDP growth [1–4]. There are a lot of valuable elements in E-waste, such as Cu, Au, Ag, Sn, Pb, Zn, In, Ta, etc., and the majority of them are included in the critical technology metal list of the EU, the USA, etc. [5–8]. The recycling of E-waste has attracted interest worldwide due to its large output, rapid growth, and high economic value [9–11].

E-waste is comprised of various kinds of end-of-life electronic equipment, and their composition and physical size vary widely. Researchers and engineers generally follow the principle of "dismantling—crushing—sorting—metallurgical separation" when treating these secondary materials [3,4,9–11]. Printed circuit boards (PCBs) are the key components in electronic equipment, and hundreds of small accessories are assembled on organic polymer boards with precision. The treatment of waste printed circuit boards (WPCBs) is the critical part of E-waste recycling. A lot of studies have been conducted to find suitable technologies for treating WPCBs, and these methods can be divided into three categories,

namely physical separation, hydrometallurgical recovery routines, and pyrometallurgical processes [12–15]. In the typical physical separation treatment, WPCBs are shredded into small bits and then separated into different groups depending on their gravity, magnetism, color, etc. Metal-rich powder and resin-rich powder could be produced [12,13]. However, all these produced materials, such as resin-rich powder, mixed electronic components, etc., are categorized as hazardous solid wastes [14,15]. Moreover, the recovery rates of precious metals in WPCBs have been proven to have negative correlations with the degrees of physical separation [16]. These disadvantages seriously limit the application of physical separation technologies in practice. Hydrometallurgical technologies usually recover valuable metals from WPCBs by the "leaching—purification—precipitation/electrowinning" methods, and they were widely used in the past due to their low technical threshold and easy process control [12,13]. However, these processes have a long flowsheet, consume a lot of leaching reagents, and only limited numbers and species of valuable metals could be recovered. Waste materials, such as waste water and leaching residues, always contain heavy metals, leaching liquor, and organic substances, and they should be treated further. WPCBs can be processed individually or jointly with the primary concentrates in Cu/Pb/Ni pyrometallurgical processes, valuable elements such as Au, Ag, Cu, Br in WPCBs can be efficiently recovered from different material streams, and toxic elements such as Hg, Cd, As, etc., can be immobilized by glassy slag [15]. The whole process realizes the unification of resource utilization and toxic materials treatment. Pyrometallurgical technologies have been widely recognized as the preferred methods in the field of WPCBs processing [15–19].

In order to achieve good recovery of valuable elements in WPCBs and reduce the formation of waste and pollutants, the principal flowsheet of "integrated materials mixing—pyrometallurgical process enrichment—hydrometallurgical separation" has been widely accepted by academic researchers and engineers [15–19]. A lot of efforts have been made to design and optimize these processes, and several mining and metallurgy groups and recycling enterprises in Europe, Asia, North America, and Oceania have established industrial-scale E-waste treatment lines [16–41]. These enterprises and smelters generally follow the principle of "collection—separation—smelting", and manual dismantling, physical separation, and metallurgical treatment cooperate and work together as integrated processes. Reusable parts, metals, plastics and some fuels could be recovered. Equipment and technologies of mechanical crushing, magnetic separation, and eddy current separation are well-developed and reliable now, and they have reached relatively high levels [21,22,24]. However, there are different technologies in treating these produced metal- and resin-rich materials, and they have different techniques and economic indicators. Umicore in Belgium [16–19], Arubis in Germany [23], Dowa group in Japan [25,26], Noranda in Canada [28], Rönnskar smelter in Sweden [33–36], and Sims metal management company in Australia [39,40] are using the "pyrometallurgical enrichment— hydrometallurgical separation" technology. WPCBs and products of physical separation processes are fed into a Cu/Pb smelter together with Cu/Pb concentrates, and precious metals and scattered metals in these materials are collected by copper matte/blister copper/lead bullion. These valuable metals can be enriched or recovered in the refinement processes of Cu and Pb. The "pyrometallurgical enrichment—hydrometallurgical separation" method has the advantages of large capacity, good adaptability, and high level of recovery. Müller-Guttenbrunn Group in Austria [20], Elden recycling company in Spain [21], Daimler Benz Ulm company in Germany [22], NEC company in Japan [24], Swiss RTec AG in Switzerland [29], and Hellatron recycling company in Italy [31] produce useful parts, ferrous metals, etc., through the "collection—dismantling—physical sorting" technology, and they provide raw materials for smelters. These recycling processes have relatively small investment and are suitable for countries or regions with small E-waste output. Sepro City metal in Canada [27] and WEEE metal company in France [30] process these sorted circuit boards with the pyrolysis technology. Valuable metals can be enriched, and part of these organic components transforms to pyrolysis gas and pyrolysis oil, which improves the recycling

value of organic parts. However, this method needs specified equipment, and its running cost is high [27,28]. Jiangxi Huagan Nerin Rare & Precious Metal Technology Co., Ltd., in China [41] has developed and industrialized the Nerin Recycling Technologies (NRT) in E-waste recycling, and other copper-containing solid waste can also be used as a feeding material. Precious metals and scattered metals are recovered from crude copper refining and dust treatment. The halogen-containing flue gas generated in smelting is treated with the "second combustion—gas quenching—activated carbon adsorption—tail gas treatment" technology, and gaseous pollutants can be efficiently controlled [41].

Material balance and heat balance (MB and HB) are important parts in developing and evaluating a metallurgical process. However, the MB and HB of these E-waste processing technologies are still not well-known, especially in industrial practices. In this study, the NRT smelting process was introduced, and its MB and HB were also studied with the METSIM software. The flowsheet and processing parameters of the NRT smelting process are introduced in Section 2, and the material and energy balances were studied based on the industrial practice data (Section 3). The obtained results are useful in optimizing the E-waste smelting process and also helpful in the related fields.

## 2. Industrial Practice of the NRT Technology

The raw materials handled in the NRT technology mainly include size-reduced WPCBs and unqualified PCBs, copper-containing sludge, waste resin powder, mixed electronic components, and leftovers generated during the PCB manufacturing process, etc. The schematic flowsheet of the NRT smelting technology is depicted in Figure 1.

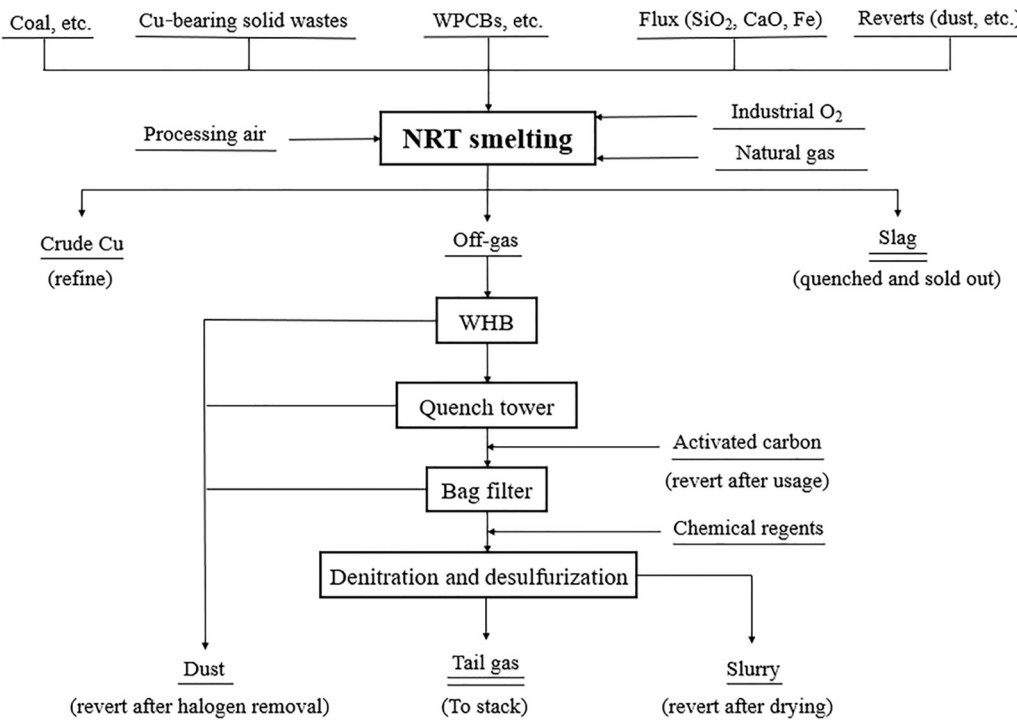

**Figure 1.** The flowsheet of E-waste processing with the NRT technology. WHB: waste heat boiler.

As shown in Figure 1, E-waste, Cu-bearing solid waste, fluxes, and fuels were added from the feeding port of the furnace by means of a belt, and processing air, industrial $O_2$, and natural gas were injected from a top blowing lance. Decomposition, oxidation, slagging, and reduction reactions took place in the furnace, and crude copper, slag, dust, and off-gas were produced. Organic components in WPCBs decomposed in the upper section of the reactor, and the products were used as fuel and reductants. Au, Ag, and other precious metals in E-waste and other raw materials were collected by the copper phase, and they could be recycled in the following refining process. Al and other impurities were

removed by slagging reactions, and the obtained glassy slag was sent to cement plants. There was a secondary combustion chamber in the gas uptake, which helped in reducing dioxin formation and improving the energy recovery of the process. Off-gas passed through the waste heat boiler and was then instantly quenched with water spraying, which reduced the amount of newly formed dioxin. As-treated gases then passed the bag filter and the denitration and desulfurization unit and lastly the vent from the stack. The plant was in operation for 585 days between January 2018 and June 2021, and the feeding materials, such as WPCBs, electroplating sludge, etc., were purchased from domestic suppliers. In total, 156,040.88 tons of raw materials (without internal reverts) were processed in that period of time, and 29,784.41 tons of crude copper and 181,975.22 tons of smelting slag were produced in that process. Chemical compositions of the feeding materials were accounted for and are presented in Table 1.

**Table 1.** Chemical compositions of the feeding materials, wt.%.

| Element | Cu | Au(g/t) | Ag(g/t) | Pt(g/t) | Pd(g/t) | Rh(g/t) | Se(g/t) | Te(g/t) | Fe |
|---|---|---|---|---|---|---|---|---|---|
| Content | 18.35 | 5.14 | 87.56 | 0.001 | 0.005 | 0.002 | 0.0004 | <0.001 | 3.30 |
| Element | Ni | Pb | Sn | Zn | Cr | Sb | Bi | As | $Al_2O_3$ |
| Content | 0.30 | 3.00 | 1.0 | 1.0 | 0.002 | 0.06 | 0.17 | 0.01 | 9.50 |
| Element | $SiO_2$ | CaO | MgO | Cl | Br | C/H/N | others | | |
| Content | 16.95 | 6.00 | 1.50 | 3.51 | 5.72 | 28.35 | 1.23 | | |

As shown in Table 1, Cu, PGMs (Au, Ag, Pt, Pd, Rh, etc.), and scattered metals (Se, Te, etc.) in E-waste are the main driving force of the recycling process, and Ni, Sn, Zn, Pb, and other valuable metals should also be recovered. The total content of halogens and organic components in the feeding materials was 37.58 wt.%, which significantly influenced the smelting process. The formation and treatment of toxic gases, such as dioxin, furan, etc., should be paid attention to. The main processing parameters of E-waste processing with the NRT technology were as follows: feeding rate of E-waste, 5.95 tons per hour; oxidation smelting duration, 3.5–4 h; reduction smelting duration, 0.5–1 h; oxygen enrichment of the gas, 21–40 vol.%; oxygen consumption, ~68.06 $Nm^3$/ton raw material; feeding rate of the processing air, ~1000 $Nm^3$/h; air leakage of the smelting process, ~1000 $Nm^3$/h; slag temperature, 1280 ± 50 °C; Fe/$SiO_2$ mass ratio in the slag, 0.8–1.4; Cu in the slag, <0.5 wt.%. Chemical compositions of the produced crude copper, waste heat boiler dust, bag filter dust, and the quenched slag were analyzed and accounted for. The results are shown in Tables 2–5.

**Table 2.** Chemical composition of crude copper, wt.%.

| Element | Cu | Au(g/t) | Ag(g/t) | Pd(g/t) | Fe | Pb | Ni | Sn | Zn | Sb | Cr | Other |
|---|---|---|---|---|---|---|---|---|---|---|---|---|
| Content | 95.32 | 13.45 | 361.89 | 0.02 | 0.15 | 0.26 | 1.14 | 1.66 | 0.11 | 0.31 | 0.035 | 1.05 |

**Table 3.** Chemical composition of the waste heat boiler (WHB) dust, wt.%.

| Element | Cu | Au(g/t) | Ag(g/t) | Pd(g/t) | Fe | Pb | Ni | Sn | Zn | Sb | Cr |
|---|---|---|---|---|---|---|---|---|---|---|---|
| Content | 14.84 | 4.18 | 300.46 | 0.008 | 6.87 | 5.38 | 1.06 | 1.88 | 9.20 | 1.08 | 0.02 |
| Element | Bi | Cd | Hg | As | Si | Ca | Al | Mg | Br | Other | |
| Content | 0.11 | 0.01 | 0.01 | 0.03 | 14.32 | 9.36 | 3.49 | 0.32 | 1.08 | 30.99 | |

**Table 4.** Chemical composition of the bag filter dust, wt.%.

| Element | Cu | Au(g/t) | Ag(g/t) | Pd(g/t) | Fe | Pb | Ni | Sn | Zn | Sb | Cr |
|---|---|---|---|---|---|---|---|---|---|---|---|
| Content | 13.54 | 3.49 | 173.62 | 0.001 | 6.12 | 4.95 | 0.06 | 1.94 | 7.35 | 0.46 | 0.05 |
| Element | Bi | Cd | Hg | As | Si | Ca | Al | Mg | Br | | Other |
| Content | 0.06 | 0.01 | 0.04 | 0.04 | 12.33 | 8.31 | 3.12 | 0.22 | 0.93 | | 40.51 |

**Table 5.** Chemical composition of the smelting slag, wt.%.

| Element | Cu | Au(g/t) | Ag(g/t) | Pd(g/t) | FeO$_x$ | SiO$_2$ | CaO | Al$_2$O$_3$ | MgO | Pb |
|---|---|---|---|---|---|---|---|---|---|---|
| Content | 0.52 | <0.001 | <0.001 | <0.001 | 32.71 | 28.96 | 17.35 | 6.83 | 2.21 | 0.05 |
| Element | Zn | Cr | Bi | Sn | As | Co | Li | Na$_2$O | Cl | Other |
| Content | 0.47 | 0.04 | 0.004 | 0.01 | <0.005 | 0.01 | 0.03 | 0.18 | 0.13 | 10.33 |

As shown in Table 2, the average grade of copper in the as-produced metal phase was 95.32 wt.%, and this phase also collected Au, Ag, Ni, Sn, Pb, Sb, Zn, and other valuable metals. As shown in Tables 3 and 4, the as-produced dusts were rich in Cu, Au, Ag, Ni, Sn, Pb, and Zn, which indicates that halogens and organic components in raw materials had obvious influence on the smelting behavior of these metals. As-produced dusts should be considered as important materials for valuable elements recovery. As shown in Table 5, the copper content in the slag was 0.52 wt.%, and the contents of Au and Ag were smaller than 0.001 g/t, which revealed that the NRT smelting process had good recovery of precious metals. The smelting process used the FeOx-SiO$_2$-CaO-Al$_2$O$_3$ slag system, and the Fe/SiO$_2$ mass ratio ranged from 0.8 to 1.4. The contents of CaO and Al$_2$O$_3$ in the slag were 17.35 wt.% and 6.83 wt.%, respectively, which is consistent with the calculated results [42].

## 3. Results and Discussions

### 3.1. Material Balance and Element Distribution Behaviors

The NRT smelting process was simulated with the METSIM software (METSIM International, LLC, 3994 E 700 N, Churubusco, IN 46723, USA) [43,44]. The composition of different raw materials, chemical reactions that took place in the process, and processing parameters optimized during the industrial practice were used to simulate the NRT smelting process (Appendices B and C). The obtained materials balance is shown in Table 6.

**Table 6.** The materials balance of E-waste smelting with the NRT technology.

| | Element | t/h | Cu | | Fe | | SiO$_2$ | | CaO | | Al$_2$O$_3$ | |
|---|---|---|---|---|---|---|---|---|---|---|---|---|
| | | | wt.% | t/h | wt.% | t/h | wt.% | t/h | wt.% | t/h | wt.% | t/h |
| Input | E-waste | 5.95 | 18.35 | 1.09 | 3.30 | 0.20 | 16.95 | 1.01 | 6.00 | 0.36 | 9.50 | 0.57 |
| | Scrap Cu | 0.80 | 92.00 | 0.74 | 3.00 | 0.02 | 0.00 | 0.00 | 0.00 | 0.00 | 0.00 | 0.00 |
| | Industrial waste | 4.46 | 14.95 | 0.67 | 28.89 | 1.29 | 31.01 | 1.38 | 19.66 | 0.88 | 1.59 | 0.07 |
| | Silica | 0.30 | 0.00 | 0.00 | 1.00 | 0.00 | 92.00 | 0.28 | 0.00 | 0.00 | 0.00 | 0.00 |
| | Limestone | 0.64 | 0.00 | 0.00 | 0.00 | 0.00 | 4.50 | 0.03 | 55.07 | 0.35 | 0.00 | 0.00 |
| | Coal | 0.50 | 0.00 | 0.00 | 0.00 | 0.00 | 5.00 | 0.03 | 3.00 | 0.02 | 0.00 | 0.00 |
| | Fe flux | 0.90 | 0.00 | 0.00 | 95.00 | 0.86 | 0.00 | 0.00 | 0.00 | 0.00 | 0.00 | 0.00 |
| | Total | 13.55 | | 2.50 | | 2.37 | | 2.73 | | 1.61 | | 0.64 |
| Output | Crude Cu | 2.39 | 95.32 | 2.28 | 0.15 | 0.00 | 0.00 | 0.00 | 0.00 | 0.00 | 0.00 | 0.00 |
| | Slag | 9.22 | 0.52 | 0.05 | 24.30 | 2.24 | 28.96 | 2.67 | 17.35 | 1.60 | 6.83 | 0.63 |
| | Dust | 1.23 | 13.60 | 0.17 | 10.33 | 0.13 | 4.88 | 0.06 | 0.87 | 0.01 | 0.91 | 0.01 |
| | Off-gas | 0.71 | 0.00 | 0.00 | 0.00 | 0.00 | 0.00 | 0.00 | 0.00 | 0.00 | 0.00 | 0.00 |
| | Total | 13.55 | | 2.50 | | 2.37 | | 2.73 | | 1.61 | | 0.64 |

As shown in Table 6, the feeding rate of E-waste was 5.95 t/h, and the copper content was only 18.35 wt.%. Scrap copper and copper-containing industrial solid waste, such as electroplating sludge, etc., were also added as feeding materials. The amount of Cu in input steams equaled that in output streams, and the comprehensive recovery of Cu reached 98%, while the Cu in dust was regarded as recycled. The Cu and Fe contents in the produced crude copper were 95.32 wt.% and 0.15 wt.%, respectively. There was a 4.53 wt.% vacancy in this phase, and the main one was dissolved oxygen. Au, Ag, Ni, Sn, Pb, Zn, etc., were also found in the crude copper phase, and they can be recycled in the Cu refining process. Silica, limestone, and Fe were used as fluxes in monitoring properties of the FeOx-SiO$_2$-CaO-Al$_2$O$_3$ slag, and the as-produced slag inherited a Fe/SiO$_2$ mass ratio of 0.84. The dust rate of the NRT process was 9%, which was higher than with the typical top-blowing technologies [45,46]. This mainly resulted from the presence of halogen elements and metals entering the gas phase in the form of halides. The smelting process was carried out in a batchwise model. E-waste, fluxes, and other raw materials were added in the oxidation smelting stage, and the processing condition of the furnace was changed to the reduction smelting stage before copper and slag tapping. There were certain amounts of organic components in the feeding materials, and these organic parts mainly underwent combustion reactions, which could provide some heat for the smelting process. Incompletely burnt materials were ignited by means of a natural gas burner in the secondary combustion chamber in the upper section of the furnace. Crude copper phase formed in the oxidation smelting process, and high copper-containing slag was also produced. The copper in the slag was recovered in the reduction smelting stage. When the copper content reached the desired value, the slag and copper were tapped. The distribution behaviors of the representative elements were calculated, and the obtained results are shown in Figure 2.

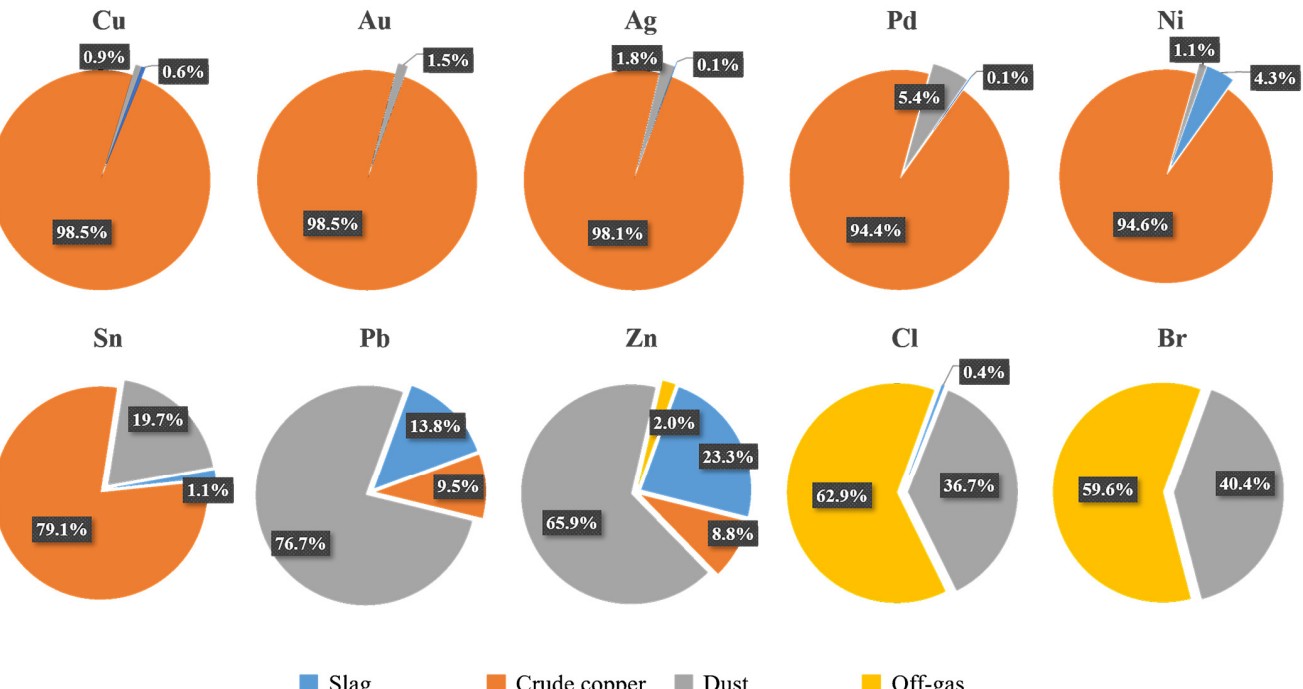

**Figure 2.** The distribution behaviors of Cu, Au, Ag, Pd, Ni, Sn, Pb, Zn, Cl, and Br.

As shown in Figure 2, the majority of these Cu, Au, Ag, and Ni in the feeding materials entered the crude copper phase, and they could be readily recovered in the Cu refining process. There were 1.5% Au, 1.8% Ag, and 5.4% Pd terminating in the dust phase, and this was caused by the reactions between these metals and halogen elements at elevated temperatures. The losses of Cu, Ag and Ni in the slag phase were 0.6%, 0.1%, and 4.3%,

respectively. Furthermore, 79.1% Sn entered the crude copper phase, and 19.7% of the total ended up in the dust phase. Thus, the recovery of Sn should come from both the Cu refining process and the dust treatment process. Up to 76.7% Pb and 65.9% Zn were enriched in the dust, and this was also caused by halogen elements. If the dust was simply returned to the smelter, there would be an accumulation of these metals. The dust rate of the smelting process would also increase. The proportions of Pb and Zn entering the slag phase were 13.8% and 23.3%, respectively, and only 9.5% Pb and 8.8% Zn ended up in the crude copper phase. Therefore, the recovery of Pb and Zn was mainly determined by the treatment of the dust. About 60% of Cl and Br entered the off-gas, and their proportions entering the dust were 36.7% and 40.4%, respectively. The distribution behaviors shown in Figure 2 show that the recovery of precious metals should come from both the crude copper refining process and the treatment of flue dust. The recycling of halogen elements can be achieved in dust treatment as well.

### 3.2. Energy Balance

The heat balance of the process was also calculated with the METSIM software, and the results are presented in Table 7.

**Table 7.** The energy balance of E-waste smelting with the NRT technology.

| Heat Input | MJ/h | % | Heat Output | MJ/h | % |
|---|---|---|---|---|---|
| Raw materials | 20 | 0.02 | Crude copper | 4080 | 5.13 |
| Fuel combustion | 26,260 | 33.04 | Smelting slag | 22,630 | 28.47 |
| Reaction enthalpy | 53,200 | 66.94 | Dust | 1830 | 2.30 |
| - | | | Off-gas | 38,440 | 48.36 |
| - | | | Loss to environment | 12,500 | 15.73 |
| Total | 79,480 | 100 | Total | 79,480 | 100 |

As shown in Table 7, chemical reaction enthalpies were the largest part in heat input and accounted for 66.94%. Fuel combustion provided about 1/3 of the total heat. The off-gas phase took away 48.36% of the total energy. The heat taken away by the smelting slag accounted for 28.47%. The results obtained from the calculation were validated with the practical data, and they were in good agreement with the statistical data. In the primary copper-making process, the burning of sulfide minerals could offer sufficient heat for the process. In the NRT smelting process, 33.04% of the total input heat was offered by fuels, which drove the process along with the heat generated from organics burning and slagging reactions. The obtained heat balance results are important in optimizing the structure of raw materials and are also useful in finding strategies in reducing carbon emission and energy consumption of the process.

## 4. Conclusions

The materials and energy balance of the E-waste smelting process with the NRT technology were studied with the METSIM software, and these optimized processing parameters and collected practical data were used in the simulation process. The main conclusions are as follows:

(1)　E-waste, together with other Cu-bearing solid waste, were processed with the NRT smelting technology, and the optimized processing parameters were as follows: feeding rate of E-waste of 5.95 t/h, slag temperature of 1280 °C, slag $Fe/SiO_2$ mass ratio of 0.8–1.4, and the CaO content in slag of 15–20 wt.%. The obtained crude copper contained 95.32 wt.% Cu, and the copper in slag was 0.5 wt.%.

(2) The smelting process was simulated with the METSIM software on the basis of these practical data, and the materials and energy balances were obtained. The total heat input of the process was 79,480 MJ/h, and the chemical reaction enthalpies accounted for 66.94% of the total. The energy from fuel combustion contributed another 1/3 of the heat input, which significantly improved the energy consumption of the NRT smelting technology. The energy was mainly brought away by the off-gas and the smelting slag, and their proportions were 48.37% and 28.47%, respectively. These results could be used in guiding and optimizing these E-waste smelting processes.

(3) Cu, Au, Ag, Pd, Ni, and Sn in the raw materials mainly entered the crude copper phase, and their proportions were 98.49%, 98.04%, 94.11%, 94.4%, 94.6%, and 74.1%, respectively. About 40% of the total halogen elements, together with 1.5% Au, 1.8% Ag, 1.1% Ni, 76.73% Pb, 67.22% Zn, and 19.7% Sn, entered the dust phase. The presence of halogen elements changed metal behaviors in an obvious manner, and the recovery of valuable metals should come from the crude copper refining process and the treatment of dusts. Halogen elements could also be recovered in the dust treatment process.

**Author Contributions:** Conceptualization, Z.L.; Project administration, L.X.; Methodology, F.Y. and L.X.; Investigation, F.Y. and L.X.; Data curation, F.Y.; Visualization, F.Y.; Supervision, Z.L.; Review, L.X.; Writing—original draft preparation, F.Y. and L.X.; Writing—review and editing, L.X.; Validation, L.X. All authors have read and agreed to the published version of the manuscript.

**Funding:** The research was funded by the National Solid Waste Project (No. 2018YFC1902503) and the National Natural Science Foundation of China (No. 52004341).

**Data Availability Statement:** Relevant data have been shown in the paper.

**Acknowledgments:** The authors acknowledge all the authors who contributed to this article.

**Conflicts of Interest:** The authors declare no conflict of interest.

## Appendix A

**Table A1.** Basic information on the typical industrial-scale E-waste treatment practice.

| No. | Company | Raw Materials Capacity | Flowsheet | Main Technique and Economic Indicators | Refs. |
|---|---|---|---|---|---|
| 1 | Umicore Group, Belgium | Waste electrical and electronic equipment (WEEE), waste automotive catalysts, etc., over 200 different types, 350 Kt per year | Base metal operations (Pb, Bi, Sb, Sn, As, In, Se, Te): blast furnace—Pb refinery—special metals refinery Precious metal operations (Cu, Ag, Au, Pt, Pd, Rh, Ir, Ru): copper smelter—leaching—electrowinning—precious metal refinement | 1. Recovery of Au, Ag, Pd, Pt, Rh, Ir, Ru, Cu, Pb, Zn, Ni, Sn, Bi, In, Se, Te, Sb, As, Co, REE. 2. $CO_2$ emission: 3.73 tons/ton metal produced (primary metal production: 17.1 tons/ton metal). | [16–19] |
| 2 | Müller-Guttenbrunn Group, Austria | WEEE, etc., 80 Kt per year | The process handles collection—depollution—shredding—ferrous metal separation—nonferrous separation—plastic recycling | 1. Recovery rate of 85% (Cu, PGs). 2. Saving of over 1 million tons of $CO_2$. | [20] |

**Table A1.** *Cont.*

| No. | Company | Raw Materials Capacity | Flowsheet | Main Technique and Economic Indicators | Refs. |
|---|---|---|---|---|---|
| 3 | Eldan Recycling, Spain | WEEE, etc., 9.5–11.5 t/h | Chopping and shredding—electromagnetic separator—eddy current separator—heavy granulator—separation table | 1. Recovery of ferrous and stainless, non-ferrous metals (Al, Cu, brass, and PCBs), refining material containing Cu, brass, Zn, Pb, PMs, etc., organic fraction with plastic, rubber, tree, textile, etc. | [21] |
| 4 | Daimler Benz, Ulm, Germany | WPCBs, etc. | Initial coarse size reduction—magnetic separation—low-temperature grinding—sieving and electrostatic separation | 1. Recovery of metal fractions (Al, Cu, brass, and PCBs) and non-ferrous metal fractions. | [22] |
| 5 | Arubis, Germany | WPCBs, scrap copper, Cu-bearing residues, etc. | Scrap Cu is processed in convertors and anode furnaces; WPCBs treated with the pyrometallurgical preparation—smelting and refining technology | 1. Recovery of Cu, Ni, Sn, Pb, and the PMs. 2. Iron silicate sand. | [23] |
| 6 | NEC Group, Japan | WPCBs | Automatic disassembly—mechanical separation—gravity and electrostatic separation | 1. Collection of waste electrical capacitors. 2. Production of Cu-rich powder and glass fiber–resin powders. | [24] |
| 7 | Dowa Group, Japan | WPCBs, residue from the Zn smelting process | The recycling network centering on Kosaka Smelting (TSL) and Refining and Akita Zinc | 1. Recovery of 17 different kinds of valuable metals. 1. Recovery of Cu, Zn, Au, Ag, Co. | [25,26] |
| 8 | Sepro Urban Metal Process, Canada | WPCBs and other e-waste materials | Crushed – thermal treatment – copper foil separation—low gravity separation—high-gravity separation | 1. Recovery of Au, Ag, Pt, Pd, Cu. | [27] |
| 9 | Noranda, Canada | WEEE, etc., 100 Kt per year | Cotreatment with copper concentrate in the Noranda smelting process | 1. Recovery of Cu, Au, Ag, Pt, Pd, Se, Te, Ni. | [28] |

**Table A1.** *Cont.*

| No. | Company | Raw Materials Capacity | Flowsheet | Main Technique and Economic Indicators | Refs. |
|---|---|---|---|---|---|
| 10 | Swiss RTec AG, Switzerland | WEEE, copper scrap, etc., 1–25 t/h | Crushing—magnetic separation—screening—eddy current separation—sensor sorting | 1. Recovery of Cu, Al, and PGMs. 2. Recovery of steel and plastics. | [29] |
| 11 | WEEE Metallica, France | WPCBs, etc., 25 Kt per year | Crushing—magnetic separation—eddy current separation—pyrolysis | 1. Recover high-grade Cu and PMs. | [30] |
| 12 | Hellatron Recycling, Italy | Solid urban waste, car batteries, solar panels, alkaline batteries, WEEE, etc. | Manual separation—size reduction—pyrolysis—gas flow classification | 1. Recovery of Cu, Al, Fe. 2. Dust with PMs. | [31] |
| 13 | Attero Recycling, India | WEEE, etc., 12 Kt per year | Crushing—manual separation—magnetic separation—eddy current separation—smelting and refining | 1. Recovery of Cu, PGMs. | [32] |
| 14 | Rönnskar Smelter, Sweden | WEEE, scrap copper, etc., 100 Kt per year | Cotreatmeant with lead concentrate in the Kaldo process—Cu and Pb refining—PM refining | 1. Recovery of Cu, Ag, Au, Pd, Ni, Se, Zn, Pb, Sb, In, Cd. | [33–36] |
| 15 | LS-NIKKO Group, South Korea | WEEE, etc. | Classification—disassembly—TSL smelting—electrolytic refining | 1. Recovery of Cu, Au, Ag, Se, Te, Pd, Pt, Ni, Bi. | [37,38] |
| 16 | Sims Metal Management Ltd., Australia | WEEE, etc., 2.5 Kt per year | Manual sorting—disassembly, crushing—TSL smelting | 1. Recovery of Cu, Pb, PGMs. | [39,40] |
| 17 | Jiangxi Huagan Nerin Precious Metal Technology Co., Ltd., China | WEEE, Cu-bearing solid waste, etc., 20 Kt per year | Manual separation—crushing—oxidation smelting—reduction smelting—refining | 1. Recovery of Cu, Au, Ag, PGMs, Pb, Zn, Sn, Br. 2. Energy consumption: < 150 kgce/ton raw material. | [41] |

**Appendix B**

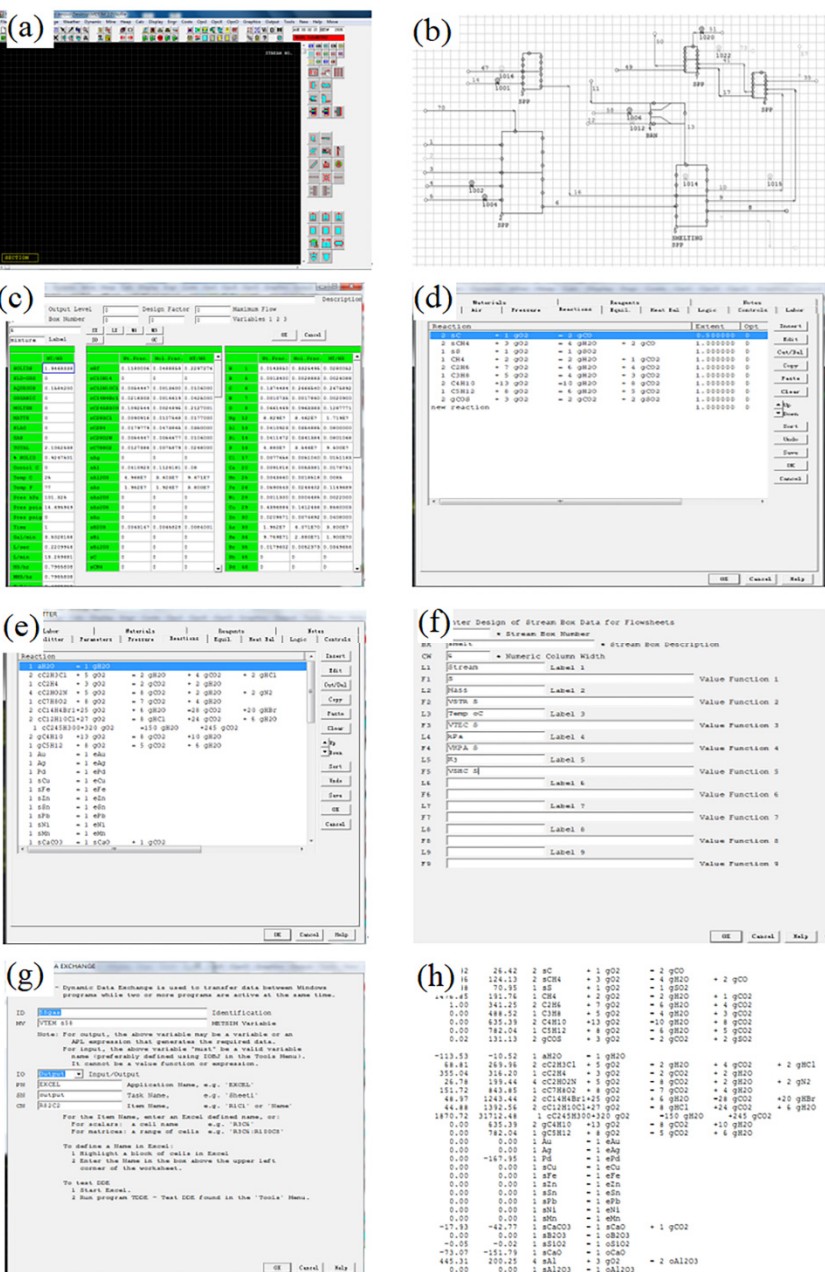

**Figure A1.** The calculation process of the MESTSIM simulation. (**a**) The main interface of METSIM, (**b**) the built-in NRT process in the software, (**c**) the input of raw materials in the calculation, (**d**) the definition of chemical reactions, (**e**) the setting of the progress of different reactions, (**f**) the setting of units of output parameters, (**g**) the definition of the displaying interface, and (**h**) the obtained energy balance.

**Appendix C**

Reactions, such as fuels and organics burning, elements slagging, etc., were input in the METSIM simulation process.

$$CH_4 \text{ (g)} + 2O_2 \text{ (g)} = 2H_2O \text{ (g)} + CO_2 \text{ (g)} \tag{A1}$$

$$2CH_4 \text{ (g)} + 3O_2 \text{ (g)} = 4H_2O\text{(g)} + 2CO \text{ (g)} \tag{A2}$$

$$C\ (s) + O_2\ (s) = CO_2\ (g) \tag{A3}$$

$$2C\ (s) + O_2\ (g) = 2CO\ (g) \tag{A4}$$

$$2CO\ (g) + O_2\ (g) = 2CO_2\ (g) \tag{A5}$$

$$4Cu\ (s) + O_2\ (g) = 2Cu_2O \tag{A6}$$

$$2Fe\ (s) + O_2\ (g) = 2FeO \tag{A7}$$

$$6FeO\ (s) + O_2\ (g) = 2Fe_3O_4 \tag{A8}$$

$$2C_2H_3Cl\ (c) + 5O_2\ (g) = 2HCl\ (g) + 4CO_2\ (g) + 2H_2O\ (g) \tag{A9}$$

$$2C_2H_4\ (c) + 6O_2\ (g) = 4CO_2\ (g) + 4H_2O\ (g) \tag{A10}$$

$$2C_7H_8O_2\ (c) + 9O_2\ (g) = 7CO_2\ (g) + 8H_2O\ (g) \tag{A11}$$

$$4CH_3NO_2\ (c) + 10O_2\ (g) = 7CO_2\ (g) + 6H_2O\ (g) + 4NO_2\ (g) \tag{A12}$$

$$2C_{15}H_{16}O_2\ (c) + 21O_2\ (g) = 15CO_2\ (g) + 16H_2O\ (g) \tag{A13}$$

$$2C_{12}H_{10}Cl\ (c) + 27O_2\ (g) = 8HCl\ (g) + 24CO_2\ (g) + 6H_2O\ (g) \tag{A14}$$

$$C_6H_5Br\ (c) + 7O_2\ (g) = 6CO_2\ (g) + 2H_2O\ (g) + HBr\ (g) \tag{A15}$$

$$2C_{14}H_4Br\ (c) + 25O_2\ (g) + 6H_2O\ (g) = 20HBr\ (g) + 28CO_2\ (g) \tag{A16}$$

$$4Fe_3O_4 + CH_4\ (g) = 12FeO + CO_2\ (g) + 2H_2O\ (g) \tag{A17}$$

$$4Cu_2O + CH_4\ (g) = 8Cu + CO_2(g) + 2H_2O\ (g) \tag{A18}$$

$$Fe_2O_3 + CO\ (g) = 2FeO + CO_2\ (g) \tag{A19}$$

$$Cu_2O + CO\ (g) = 2Cu + CO_2\ (g) \tag{A20}$$

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
