# Peer review of "Materials and Energy Balance of E-Waste Smelting—An Industrial Case Study in China"

_metals, doi:10.3390/met11111814_

Round 1
Reviewer 1 Report
This paper is useful to know the materials and energy balance of E-waste in China and also it is well added for other companies in other countries.
Please check the following things.
Do you have any meanings more than the significant values of 4 in Table 7.?
79476 MJ/h, etc. →79480?
Line 35 (p.1) Pd is also add because Pd is important for recent valuable values.
Also, please write the Pd percentage by analysis in Table 2,3,4,5.
Because Pd is shown in Table 1.
At the first appearance of abbreviation such as WPCBs, show the whole words.
Waste printed circuit boards (WPCBs)
WHB in Figure1.
For METSIM software, please write the produced company’s name in the sentence, too.
p.4 Nm3/h, Write as superscript.
There are two conclusions. How about change the name “5. Conclusions” like conclusion for discussion.
Reviewer 2 Report
Although the paper addresses a very interesting and appealing research topic, it presents clear failings detailed in the following list:
Abstract:
- The methodology adopted to achieve the target of the research work is not detailed; the authors provide information related to software adopted for evaluation (i.e., METISM). No further details on the analytical approach as well as on functions evaluated have been provided. At the same time, let me suggest reducing, in summary, the description of the results achieved.
- Please define the acronyms’ meaning when they are introducing the first time, in summary, and in the body of the manuscript.
Introduction:
- The target of the paper is not well stated.
- The manuscript's overall contribution needs to be better explored: what exactly is the key finding of this paper? Which research questions are faced by the study? To this end, the discussion needs to build explicit links to particular literature that is challenged, extended, or otherwise influenced by this study.
- Please provide a brief description of the paper's structure (e.g., in section 2 … is provided, in section 3,… etc.)
Materials and methods:
- Please provide more information on the feeding materials provided between Jan 2018 and Jun 2021, summarized in table 1 (e.g. which processes are referred? Global, national or local, scale?, one or more plant/s?, etc.).
- Please describe the methodology adopted by METSIM software. For instance, how the materials balance is evaluated by software, which algorithm is adopted, which are the objective functions considered, how changes the balance adopting other methodologies, etc. This part is completely missing in the manuscript. In my view, the authors consider the software as a ‘black-box’, where has been introduced e-waste features and the corresponding materials balance is estimated… no consideration on the approach adopted was detailed in the research work.
Results’ discussion:
- No discussion on the results achieved was included in the paper.
Conclusion:
- A synthesis of major gaps filled by the manuscript and further development of the approach used could help to focus the results of the scientific investigation carried out.
Mistakes:
- The content of sections 4 and 5 are inconsistent with the rest of the work.
Reviewer 3 Report
Materials and energy balance of E-wastes smelting---An industrial case study in China is very interesting paper. Some improvement is required.
Abstract:
The rare earth elements are not considered in electronic waste. Why?
Line 46: WPCBs, (waste printed circuit boards)- please to write full name only one time.
Line 120: What i s the meaning of WHB. Can you explain it in text below the Figure 1.
LIne 154: Chemical compositions of the WHB dust, wt.%. Please to give a full name of WHB.
Line 155: How to explain the high amount of copper in bag filter dust (Table 4. Chemical compositions of the bag filter dust, wt.%. )
Line 246: This section (conclusion) is mandatory and shell be added to the manuscript. Please to give an estimation of possible validation of Materials and energy balance of E-wastes smelting using the application of Nerin Recycling Technologies (NRT)
Round 2
Reviewer 2 Report
Dear authors, the revisions adopted have improved the work.
Nevertheless, the following issues should be evaluated before final acceptance:
- Please state in the manuscript the target of the paper as provided in your reply to the reviewer.
- Please provide the manuscript's overall contribution with explicit links to particular literature that is challenged, extended, or otherwise influenced by this study. In other words, which are the limits of existing studies on this topic, and how does your research work face them.
- Please change the manuscript's structure, including a section “Discussion and Result”, in my view, the current sections 2.2 and 2.3 can be re-arranged consistent with this organization of the manuscript.
- Please improve the layout of distributions shown in fig. 2. The numbers are very hard to read.
- Please check the lines from 244-251; the format style is in bold.
- Please check the alignment of the text included in the conclusion section
- In the first part of the conclusion section, please summarize the scope of the research work conducted. Moreover, please avoid starting a section with a bulleted/numbered list.
- Please include in the conclusion section the further developments of the approach adopted.
- Please evaluate the possibility to include the MESTSIM screenshot and/or reactions (provided in your last reply to the reviewer) in the appendix or other section of the manuscript. In my view, this information could improve the clarity of the approach adopted.
Round 3
Reviewer 2 Report
The revisions adopted have improved the work. The suggestions proposed in the review report are included in the last draft of the paper. The major gaps filled by the manuscript are now described in “Introduction”, and “Results”. The mistakes identified in the previous draft of the paper have been corrected.